# Optimizing Water, Temperature, and Density Conditions for In Vitro Pea (*Pisum sativum* L.) Germination

**DOI:** 10.3390/plants13192776

**Published:** 2024-10-03

**Authors:** Zoltán Kende, Petra Piroska, Gabriella Erzsébet Szemők, Hussein Khaeim, Asma Haj Sghaier, Csaba Gyuricza, Ákos Tarnawa

**Affiliations:** 1Institute of Agronomy, Hungarian University of Agriculture and Life Sciences, Páter Károly u.1, Gödöllő, 2100 Pest, Hungary; 2Field Crops Department, College of Agriculture, University of Al-Qadisiyah, Al Diwaniyah 58002, Iraq

**Keywords:** pea germination, seed density, temperature sensitivity, antifungal treatments, germination metrics

## Abstract

This study aimed to determine the optimal water, temperature, and density conditions, alongside antifungal treatments, for pea (*Pisum sativum* L.) germination in a laboratory setting, with implications for research, breeding, and microgreen production. Germination and early seedling growth were assessed across various temperatures (5 °C to 40 °C), water levels (0–14 mL per Petri dish), seed densities (5, 7, 9, and 11 seeds per Petri dish), and antifungal treatments (Hypo and Bordeaux mixture). The results indicated that optimal germination occurred between 15 °C and 25 °C, with peak performance at 25 °C. Water levels between 7 and 11 mL per 9 cm diameter Petri dish supported robust root and shoot development, while minimal water levels initiated germination but did not sustain growth. Five seeds per Petri dish was optimal for healthy development, whereas higher densities led to increased competition and variable outcomes. Antifungal treatments showed slight improvements in germination and growth, though differences were not statistically significant compared to controls. The study’s novelty lies in its holistic approach to evaluating multiple factors affecting pea germination, offering practical guidelines for enhancing germination rates and seedling vigor. These findings support efficient and resilient crop production systems adaptable to varying environmental conditions, contributing to sustainable agriculture and food security. Future research should explore these factors in field settings and across different pea cultivars to validate and refine the recommendations.

## 1. Introduction

*Pisum sativum* L., a member of the Fabaceae family, is a widely cultivated legume of significant nutritional and economic importance, originating from the Mediterranean basin and Central Asia [1]. Peas have been a staple in human diets for centuries and are now grown globally, including in temperate and subtropical regions [2]. In 2022, approximately 7.2 million hectares of peas were cultivated worldwide, with major producers including Canada, China, France, and Russia, and substantial but lower-yielding areas in countries like Pakistan and Ethiopia (FAOSTAT). Global pea production, amounting to about 18 million tons, highlights their adaptability and essential role in sustainable agriculture [1]. Rich in proteins, essential amino acids, dietary fibers, vitamins, minerals, and phytochemicals, peas offer numerous health benefits [2]. Consumed fresh, frozen, canned, or in processed foods, their low fat content and low glycemic index contribute to their nutritional value [3]. Additionally, peas play a crucial role in sustainable farming by fixing atmospheric nitrogen through symbiosis with rhizobia, enriching soil fertility, and reducing the need for synthetic fertilizers [4,5,6].

Advancements in genetic and genomic research have led to improvements in pea cultivars, aiming to boost yield, nutritional quality, and resistance to biotic and abiotic stresses [7]. However, pea production still faces challenges such as yield instability and susceptibility to diseases and environmental stresses. Addressing these challenges through innovative breeding techniques and sustainable farming practices is essential for the stability and growth of global pea production [8]. This is especially true in the case of the germination of peas. During imbibition, the first step of germination, seeds rapidly absorb water, allowing the seed coat to expand and soften under optimal temperatures [9]. This rehydration activates internal physiological processes, including a large increase in respiration, leading to the emergence of radicles and plumules as the seed coat cracks [10]. Germination involves systematic physiological and morphogenetic processes, including energy transfer, nutrient translocation from the endosperm, and metabolic changes [11]. Key signals of germination include the reactivation of transcription, translation, and DNA repair, followed by cell elongation and division [12]. Hydrogen peroxide (H_2_O_2_) has been identified as a signaling molecule that can enhance germination and seedling growth through specific changes and help inhibit stressors at proteomic, transcriptomic, and hormonal levels [10,13]. Proteomic analysis has also revealed significant changes in protein expression during germination, highlighting the importance of metabolic reorganization and activation of protective systems [12]. Furthermore, mitochondrial function plays a crucial role in energy supply during germination, with impaired mitochondrial performance leading to reduced ATP production and seedling growth [14]. Environmental factors such as temperature also significantly influence germination patterns and seedling establishment, with variations in seed coat thickness and proanthocyanidin content affecting dormancy and germination rates [15].

The COVID-19 pandemic has highlighted the importance of food security in the face of accelerating climate change. Pea production is ideal for urban and peri-urban food security due to its nutrient-rich content and short growth cycle, making it perfect for microgreens. Pea microgreens, packed with nutrients and antioxidants, grow quickly and are well-suited for limited spaces [16,17]. Also, its production aligns with the European Green Deal’s goals of reducing chemical pesticide and fertilizer use by promoting organic farming and integrated pest management [18,19]. Pea plants improve soil health through nitrogen fixation, reducing reliance on synthetic fertilizers and enhancing soil biology [20,21].

Optimal germination conditions for pea seeds are crucial for crop production, microgreens, and research, especially when resources are limited. Ensuring successful germination, particularly under abiotic stress like drought and extreme temperatures, is key to achieving reliable and efficient outcomes. This knowledge supports resilient food systems capable of withstanding climate change, thereby contributing to sustainable food security [22].

This study aims to dissect these intricacies through a series of targeted objectives. This study aimed to assess the ability of pea seeds to germinate at different moisture and temperature levels. Our goal was also to identify the optimal number of seeds per Petri dish, with a 9 cm diameter for laboratory experiments, and to establish an effective antifungal treatment strategy. To address these pressing concerns, our research focuses on four key objectives, each designed to unravel aspects of pea germination and growth critical for enhancing agricultural resilience and productivity. Therefore, our inquiry addressed four key aims:Investigate the impact of temperature on pea seedling germination, growth, and duration, filling a knowledge gap on climate change’s effects and identifying optimal temperature thresholds.Determine the minimum, optimal, and maximum water levels for germination using the thousand-kernel weight (TKW) base, providing insights for water-efficient farming in arid regions.Explore the effects of seed number and seedling density on germination and lid uncovering, developing effective seeding strategies for both lab and large-scale agriculture.Examine how seed priming and antifungal treatments enhance seedling health and vigor, reducing losses from soilborne pathogens and contamination.

The aim of this study was to determine the optimal water, temperature, and density conditions for pea (*Pisum sativum* L.) germination in laboratory conditions, with implications for research, breeding, and microgreen production. Continuing our research series on crops like maize [23], wheat [24], barley [25], sunflower [26], and rapeseed [27], this study on pea germination delves into optimizing germination amidst environmental stresses. It focuses on the effects of temperature, moisture, seed priming, and antifungal treatments to boost pea germination in laboratories and fields. However, our goal was not to test germination under field conditions, in particular because germination is extremely crucial in laboratory tests where only a small amount of seeds are available, and in microgreen production, where optimization is key for energy conservation. Our findings aim to provide actionable insights for researchers, breeders, and farmers.

## 2. Results

### 2.1. Germination Test

The germination of pea seeds (*Pisum sativum* L.) was evaluated at various temperatures (from 5 to 40 °C) to understand how temperature influences the germination process. The experiment utilized key metrics based on the Four-parameter hill function (Figure 1). Additionally, the area under the curve (AUC) was calculated to provide a comprehensive assessment of germination performance (Figure 2).

At 5 °C, germination started slowly, with seeds beginning to germinate around day 5. The TMGR was 7.11 days, reflecting the extended period required for seeds to reach maximum germination rates at 77.77%. The t_50_ was 7.86 days, indicating slower overall germination. The MGT was 7.90 days, and the uniformity (U) was 3.97, respectively. The AUC was 505.98, reflecting the slow and staggered germination process at this low temperature. At 10 °C, the germination rate improved compared to 5 °C. Seeds began to germinate around day 3, with the TMGR reduced to 3.15 days, and t50 reduced to 3.64 days, showing quicker germination. The MGT was 4.46 days, indicating that seeds germinated more uniformly and quickly. The uniformity (U) was 4.51 days, and the AUC was 992.96, suggesting a better overall germination performance than at 5 °C with a 100% germination rate. The germination at 15 °C was significantly faster and more uniform. Seeds began to germinate around day 2, with a TMGR of 2.07 days and t_50_ of 2.53 days. The MGT was 3.33 days, and the uniformity (U) was 3.57 days, indicating a more synchronized germination process with a 97.78% germination rate. The AUC was 1085.71, reflecting the efficient germination at this temperature.

At 20 °C, germination started to be optimal with a germination rate of 97.8%. Seeds started germinating around day 2, with a TMGR of 1.87 days and t_50_ of 1.88 days. The MGT was further reduced to 2.38 days, and the uniformity (U) decreased to 0.22 days, indicating highly uniform germination. The AUC was 1184.72, reflecting the peak performance at this temperature. Germination at 25 °C continued to be highly efficient, with seeds beginning to germinate around day 2 also. The TMGR was 1.86 days, and t_50_ was 1.86 days. The MGT was 2.36 days, and the uniformity (U) remained high at 0.22 days as well. The AUC was 1213.48, indicating excellent germination performance similar to 20 °C with a 100% germination rate.

At 30 °C, germination remained efficient but showed the first signs of thermal stress. Seeds began germinating quickly, with a TMGR of 1.83 days with a t_50_ of 1.84 days. The MGT was 2.34 days, and the uniformity (U) remained at 0.22 days, indicating a plateau in performance compared to 20 °C and 25 °C. The AUC was 1215.81. At 35 °C, germination showed further signs of thermal stress with a 95.5% successful germination rate. Germination started quickly, with a TMGR of 1.66 days and t_50_ of 1.72 days. The MGT was 2.24 days, and the uniformity (U) decreased significantly to 0.83 days. The AUC was 1171.78, reflecting the onset of stress affecting germination efficiency. At 40 °C, germination performance dropped significantly with a germination rate of 2.22%. Germination began around day 2, but the TMGR was 1.41 days, and the t_50_ data were not available because the germination stopped after day 2. The MGT was 1.92 days, and the uniformity (U) was the lowest at 0.20 days, showing highly staggered germination. The AUC was very low at 27.96, reflecting poor overall germination performance due to thermal stress.

Our results indicate that temperature significantly affects the germination characteristics of pea seeds. Optimal germination occurs between 20 °C and 30 °C, where seeds exhibit rapid, vigorous, and uniform germination. In contrast, extreme temperatures (5 °C and 40 °C) result in delayed, less vigorous, and highly variable germination. These findings underscore the importance of maintaining appropriate temperature conditions to ensure successful and uniform pea seed germination.

The germination analysis of pea seeds at different temperatures reveals significant impacts on germination characteristics, based on Figure 2. The mean germination time (MGT) (Figure 2A), which indicates that seeds germinate fastest at higher temperatures, shows—based on the Kruskal–Wallis test—that there are several confirmed significant differences in MGT across temperatures (χ^2^(2) = 30.74, *p* < 0.001). Post-hoc Dunn–Bonferroni tests revealed substantial differences between several temperature pairs (Table 1). At 20 °C, there is a notable decrease in germination time compared to 5 °C (−23.20 *), 10 °C (17.40 *), and 15 °C (14). This trend of significantly shorter germination times continues at higher temperatures. For instance, at 25 °C, the germination time is significantly reduced compared to 5 °C (−24.90 *), 10 °C (19.10 *), and 15 °C (15.70 *) but shows only a slight difference when compared to 20 °C (1.7). Similarly, at 30 °C, there are significant differences with 5 °C (−25.90 *), 10 °C (20.10 *), and 15 °C (16.70 *), with minimal variation from 20 °C (2.7) and 25 °C (1). At 35 °C, the significant difference is noted only with 5 °C (−16.90 *), while differences with higher temperatures (10 °C to 30 °C) are less pronounced. Finally, at 40 °C, the germination time significantly decreases compared to 5 °C (−28.50 *), with substantial differences from 10 °C (22.7) and 15 °C (19.3) but minimal changes compared to higher temperatures (20 °C to 35 °C). These data indicate that higher temperatures generally result in shorter pea germination times, with the most significant differences observed at the extremes (5 °C and 40 °C).

Another chosen index was Timsons’ Index (GR) (Figure 2B), which reflects germination energy and overall vigor. The Kruskal–Wallis test demonstrated significant differences in the Timson Index across the various temperatures (χ^2^(2) = 35.17, *p* < 0.001). It showed the highest values between 15 °C and 35 °C, peaking at 25 °C with a mean ± std. dev of 1180 ± 60. The index was lower at extreme temperatures, recorded at 500 ± 50 at 5 °C and 200 ± 50 at 40 °C. The post hoc analysis of Timsons’ Germination Index across various temperatures highlights significant differences, marked by asterisks. At 20 °C, there is a substantial increase in the germination index compared to 5 °C (22.70 *), 10 °C (−16.70 *), and 15 °C (−12.5). This pattern of significant increases continues at higher temperatures. For instance, at 25 °C, the germination index significantly rises compared to 5 °C (24.80 *), 10 °C (−18.80 *), and 15 °C (−14.6), with a smaller difference when compared to 20 °C (−2.1). Similarly, at 30 °C, significant increases are observed at 5 °C (25.60 *), 10 °C (−19.60 *), and 15 °C (−15.4), with minor changes from 20 °C (−2.9) and 25 °C (−0.8). At 35 °C, significant increases are noted only with 5 °C (15.70 *), while differences with higher temperatures (10 °C to 30 °C) are less pronounced. At 40 °C, the germination index significantly decreases compared to 5 °C (−5) but shows substantial increases compared to 10 °C (11) and 15 °C (15.2), and it shows higher increases from 20 °C (27.70 *), 25 °C (29.80 *), and 30 °C (30.60 *), with a marked difference from 35 °C (20.70 *). These data indicate that higher temperatures generally result in higher Timsons’ germination indices, with the most significant differences occurring at the extremes (5 °C and 40 °C) (Table 1).

The Germination Uncertainty index (Z-index) (Figure 2C), which quantifies the variability and uniformity of germination, was lowest at optimal temperatures, indicating more uniform germination. The Kruskal–Wallis test indicated significant differences in Germination Uncertainty across temperatures (χ^2^(2) = 28.55, *p* < 0.001). Based on the Dunn–Bonferroni post hoc test of the synchronization index, significant differences were observed across various temperatures. At 20 °C, there is a notable decrease in the synchronization index compared to 5 °C (−20), with increases relative to 10 °C (17.4) and 15 °C (18.3). This trend of significant differences continues at higher temperatures. For instance, at 25 °C, the synchronization index significantly decreases compared to 5 °C (−21.90 *), with increases relative to 10 °C (19.3) and 15 °C (20.2), and a slight difference when compared to 20 °C (1.9). Similarly, at 30 °C, significant decreases are observed at 5 °C (−23.00 *), while there are increases relative to 10 °C (20.4) and 15 °C (21.3), with minimal changes from 20 °C (3) and 25 °C (1.1). At 35 °C, significant decreases are noted at 5 °C (−13), with moderate increases relative to 10 °C (10.4) and 15 °C (11.3) but notable decreases compared to 20 °C (−7), 25 °C (−8.9), and 30 °C (−10). At 40 °C, the synchronization index significantly decreases compared to 5 °C (−26.60 *), with substantial increases relative to 10 °C (24.00 *) and 15 °C (24.90 *) and higher increases compared to 20 °C (6.6), 25 °C (4.7), and 30 °C (3.6), with a marked difference from 35 °C (13.6). These data indicate that higher temperatures generally result in higher synchronization indices, with the most significant differences observed at the extremes (5 °C and 40 °C) (Table 1).

Our germination experiment results show that temperature significantly impacts the germination characteristics of pea seeds. Optimal germination occurs between 20 °C and 30 °C, where seeds exhibit the shortest mean germination time, highest germination energy (Timsons’ Index), and lowest Germination Uncertainty, indicating rapid, vigorous, and uniform germination. In contrast, extreme temperatures (5 °C and 40 °C) result in poor germination performance, characterized by longer mean germination times, lower germination energy, and higher variability. These findings highlight the importance of maintaining appropriate temperature conditions to ensure successful and uniform germination of pea seeds.

### 2.2. Initial Development Test

In this study, the point of appropriate measurement for evaluating the initial development of pea seeds was when 80% of the seeds had a plumule length of 0.5 cm. According to the results, the most efficient germination occurred at temperatures of 15 °C, 20 °C, and 25 °C (Figure 3). After 10 days at 5 °C, measurable results were observed in the seeds. At 10 °C, the seedlings became measurable from the fourth day. When maintained at 15 °C, the plant parts could be accurately measured from the second day. Efficient measurements of the plant parts could also be taken from the second day at 20 °C, and at the end of the experiment, this temperature resulted in the longest seedlings. At 25 °C, the most significant results were obtained on days 8–10. At 30 °C, the seedlings did not grow more than 3 cm. At 35 °C, however, the seeds germinated but did not grow the radicle over the 0.5 cm limit, and at 40 °C, none of the seeds germinated; thus, we did not show both of them on the next graphs.

The radicle length, averaged over all measurements and temperatures, demonstrates that optimal growth occurred at 15 °C, 20 °C, and 25 °C. The mean radicle lengths at these temperatures were 8.48 ± 4.72 cm (min: 0.56 cm, max: 16.83 cm), 8.07 ± 4.90 cm (min: 0.71 cm, max: 17.78 cm), and 7.36 ± 5.03 cm (min: 0.67 cm, max: 17.33 cm), respectively. These results indicate that radicles can sustain growth across a range of temperatures, with notable growth even at higher temperatures, but growth significantly decreases at 5 °C and 30 °C. Specifically, the radicle length at 5 °C was 2.40 ± 1.18 cm (min: 0.0 cm, max: 5.94 cm), and at 30 °C, it was 3.37 ± 2.27 cm (min: 0.0 cm, max: 8.33 cm). The figures demonstrate that radicles can grow at a temperature as high as 30 °C, but the shoots only exhibit marginal growth at this temperature.

The shoot lengths also showed optimal growth at 15 °C, 20 °C, and 25 °C, with mean shoot lengths of 3.84 ± 2.89 cm (min: 0.0 cm, max: 9.22 cm), 4.02 ± 4.02 cm (min: 0.0 cm, max: 12.17 cm), and 3.91 ± 3.77 cm (min: 0.67 cm, max: 12.0 cm), respectively. At lower and higher extremes, the growth was significantly reduced. At 5 °C, the mean shoot length was 0.18 ± 0.49 cm (min: 0.0 cm, max: 3.22 cm), while at 30 °C, it was 1.51 ± 1.51 cm (min: 0.0 cm, max: 4.78 cm). Their growth was poor at 5 °C and marginal at 30 °C (Figure 4).

The seedling length (Figure 5), which combines both radicle and shoot lengths, further highlights the optimal growth temperatures. As shown in the analysis, seedlings exhibited the most significant growth at 15 °C, 20 °C, and 25 °C. The mean seedling lengths at these temperatures were 12.32 ± 7.51 cm (min: 0.56 cm, max: 26.0 cm), 12.09 ± 8.80 cm (min: 0.71 cm, max: 27.39 cm), and 11.26 ± 8.61 cm (min: 0.67 cm, max: 24.33 cm), respectively. At lower temperatures of 5 °C, the mean seedling length was only 2.58 ± 1.41 cm (min: 0.0 cm, max: 5.94 cm), and at 30 °C, it was 4.88 ± 3.64 cm (min: 0.0 cm, max: 13.33 cm).

The Mann–Whitney U test results indicate significant differences in seedling lengths between various temperature pairs (Table 2). Specifically, seedling lengths at 5 °C were significantly different from those at all other temperatures (*p* < 0.05). Significant differences were also observed between 10 °C and all other temperatures except for 15 °C (*p* = 0.886) and 20 °C (*p* = 0.333). There were no significant differences in seedling lengths between 15 °C and 20 °C (*p* = 0.886), 15 °C and 25 °C (*p* = 0.565), and 20 °C and 25 °C (*p* = 0.333), indicating these temperatures are statistically similar in their effects on growth. However, 15 °C and 20 °C showed significant differences from 5 °C, 10 °C, and 30 °C. Meanwhile, 30 °C exhibited significant differences from all other temperatures except 20 °C (*p* = 0.333).

Based on the initial development test, the most effective germination and growth of pea seedlings occurred at temperatures of 15 °C, 20 °C, and 25 °C based on their higher mean seedling lengths and the lack of significant differences among them. These temperatures supported the robust development of radicles, shoots, and overall seedling length. Temperatures outside this range, particularly at 5 °C and 30 °C, resulted in significantly reduced growth, while at 35 and 40 °C the rate of development was negligible in the initial stage.

### 2.3. Water Amount Experiment

This study investigates the impact of varying water quantities on the growth and dry weight accumulation of pea seedlings, shoots, and radicles, with water levels ranging from 0 mL to 14 mL at one milliliter intervals. The polynomial regression trends indicate strong fits with R^2^ values of 0.79, 0.92, and 0.88 for radicle, shoot, and seedling growth, respectively, and R^2^ values of 0.35, 0.66, and 0.57 for their respective dry weights, further corroborating the observed trends (Figure 6 and Figure 7). The germination rate showed a marked dependence on water levels, with no germination at 0 mL, a 2.2% germination rate at 1 mL, and significant increases observed between 8 mL and 14 mL, where rates ranged from 93.3% to 100% (97.8 ± 5.0 at 8 mL, 100.0 ± 0.0 at 14 mL). Radicle growth increased significantly with water levels, peaking around 10 mL to 12 mL at approximately 14 cm (13.8 ± 1.0 cm at 10 mL, 14.0 ± 1.0 cm at 12 mL). Tukey’s test indicated that water levels of 8 mL and above resulted in significantly higher radicle lengths compared to lower water levels (*p* < 0.05). Similarly, shoot length peaked around 12 mL at about 17 cm (17.1 ± 0.8 cm at 12 mL), with significant increases noted from 6 mL upwards. Seedling growth was most pronounced between 12 mL and 14 mL, reaching approximately 20 cm (20.2 ± 1.2 cm at 14 mL), with significant differences observed between water levels of 8 mL and higher. Dry weight accumulation also varied with water levels. Radicle dry weight peaked at 7 mL around 0.1 g (0.10 ± 0.01 g), then slightly decreased at higher levels, with significant differences noted between 7 mL and higher water levels. Shoot dry weight reached a maximum of around 12 mL at about 0.2 g (0.20 ± 0.02 g), showing significant increases from 6 mL upwards. Seedling dry weight peaked between 8 mL and 10 mL at about 0.2 g (0.20 ± 0.02 g), with significant differences observed from 8 mL upwards. These findings suggest that the optimal water potential for pea growth and dry weight accumulation lies between 8 mL and 14 mL, with even minimal levels like 2 mL sufficient to initiate germination. However, the optimal range for achieving the highest dry mass and germination percentage is between 6.95 mL and 11.6 mL. These results provide valuable insights for optimizing water usage in pea cultivation to enhance growth efficiency and yield (Table 3).

We investigated the impact of varying water quantities, measured as multiples of the thousand-kernel weight (TKW), on the growth and dry weight accumulation of radicles, shoots, seedlings, and the total dry weight (Table 4). The water treatments ranged from 0.75 mL to 14.70 mL, corresponding to TKW percentages from 16.17% to 316.84. The objective was to determine how much of the thousand-seed mass is required for optimal germination and growth (Figure 8).

Radicle dry weight showed a relatively weak correlation with TKW, with an R^2^ value of 0.12. The highest radicle dry weight was observed at lower TKW percentages, with the dry weight peaking at approximately 0.1 g at 2.30 mL (0.072 ± 0.005 g) and then gradually decreasing as the TKW-corrected water amount was increased. The shoot dry weight exhibited a stronger correlation with TKW-corrected water amount, with an R^2^ value of 0.43. The mean shoot dry weight varied from 0.0 g at very low TKW percentages to approximately 0.2 g at higher TKW percentages, with the highest shoot dry weight observed at 4.65 mL (0.158 ± 0.208 g). These data demonstrated that shoot dry weight increased with increasing TKW percentages, indicating a positive relationship between the two variables. Seedling dry weight also showed a moderate correlation with TKW-corrected water amount, with an R^2^ value of 0.31. The dry weight of seedlings ranged from 0.0 g at lower TKW percentages to about 0.258 g at higher TKW percentages (0.258 ± 0.210 g at 4.65 mL). This trend suggested that seedling dry weight increased with increasing TKW percentages, similar to the pattern observed for shoot dry weight.

The total dry weight, which combines the dry weights of radicles, shoots, and seedlings, showed an R^2^ value of 0.30, indicating a moderate correlation with TKW-corrected water amount. The mean total dry weight ranged from 0.0 g to approximately 0.258 g across the corrected water amount spectrum (0.258 ± 0.210 g at 4.65 mL) (Figure 9). This trend suggests that higher TKW percentages are associated with greater total dry weights, highlighting the importance of optimal water levels to achieve maximum dry weight accumulation.

In summary, this analysis indicates that applying water as a multiple of the TKW significantly influences the dry weights of radicles, shoots, seedlings, and the total dry weight. Higher TKW percentages generally correspond to increased dry weights, emphasizing the critical role of water management in optimizing TKW and enhancing the overall growth and yield of pea plants. The findings underscore the importance of determining the optimal amount of water, based on TKW, required for maximal germination and growth efficiency in pea production.

### 2.4. Density Experiment

In this sub-experiment, we investigated the effect of seed density on the germination and seedling development of pea seeds using different seed densities in 9 cm diameter Petri dishes, each containing 9 milliliters of water. The densities tested were five, seven, nine, and eleven seeds per Petri dish. Our objective was to determine if varying seed densities had any significant impact on the germination rates and subsequent seedling growth.

The statistical analysis of variance revealed no significant differences in the germination percentages across the different seed densities (Table 5). Specifically, the germination characteristics evaluated included inactive seeds, initial growth, radicle-only seedlings, short seedlings, normal seedlings, and aggregated germination values. For instance, the proportion of inactive seeds was low across all treatments, with mean values of 0.000 ± 0.000 for five seeds, 0.014 ± 0.045 for seven seeds, 0.000 ± 0.000 for nine seeds, and 0.009 ± 0.029 for eleven seeds. Similarly, the aggregated germination values, which represent the combined outcome of all germination categories, showed minimal variation, with values of 0.960 ± 0.127 for five seeds, 0.861 ± 0.241 for seven seeds, 0.838 ± 0.271 for nine seeds, and 0.652 ± 0.327 for eleven seeds.

Our findings suggest that the placement of five pea seeds per Petri dish consistently resulted in the growth of normal-sized shoots and roots, indicating optimal germination conditions. In contrast, higher seed densities (seven, nine, and eleven seeds per dish) exhibited a mix of germination outcomes, including ungerminated seeds, seeds with only initial growth, radicle-only seedlings, and seeds with short or normal seedlings. These mixed results were consistent across the higher seed densities, suggesting that increased competition for resources may influence the uniformity of seed germination and seedling development. Consequently, while higher densities do not significantly affect the overall germination percentage, they may lead to variability in seedling growth characteristics.

### 2.5. Antifungal Experiment

According to the data, the germination percentage was highest in the Hypo treatment (100.0 ± 0.0), followed by the Bordeaux solution (97.78 ± 4.97) and control (93.33 ± 14.91). The radicle length was also greatest in the Hypo treatment (9.88 ± 1.02 cm), indicating a significant positive effect on root growth compared to the Bordeaux solution (5.97 ± 1.29 cm) and control (9.11 ± 5.27 cm). The shoot length was the highest in the Hypo treatment (5.24 ± 1.51 cm), followed by the control (4.89 ± 2.83 cm) and Bordeaux solution (3.24 ± 1.86 cm). The overall seedling length was significantly greater in the Hypo treatment (15.12 ± 2.37 cm) compared to the Bordeaux solution (9.2 ± 2.86 cm) and control (14.0 ± 7.95 cm). The LSD values indicated that differences between treatments were not statistically significant (NS) for all measured parameters, suggesting variability within treatments (Table 6).

The germination percentage was consistently high across most antifungal treatments, with the 1 ppm, 10 ppm, 100 ppm, and 1000 ppm treatments showing similar germination rates (95.56 ± 9.94 to 100.0 ± 0.0), and slightly lower in the 10,000 ppm treatment (84.45 ± 14.91 cm). The radicle length was the highest in the 1 ppm treatment (9.57 ± 0.95 cm) and control (9.11 ± 5.27 cm), with significant reductions observed at higher concentrations, particularly in the 10,000 ppm treatment (1.77 ± 0.66 cm). The shoot length followed a similar trend, being highest in the 1 ppm treatment (7.49 ± 2.23 cm) and control (4.89 ± 2.83 cm) and lowest in the 100 ppm and 1000 ppm treatments (1.02 ± 1.33 cm and 0.92 ± 1.23 cm, respectively). The seedling length was significantly greater in the 1 ppm treatment (17.06 ± 1.48 cm) compared to other treatments, with the lowest lengths observed in the 100 ppm, 1000 ppm, and 10,000 ppm treatments (4.51 ± 2.09 cm, 4.54 ± 1.29 cm, and 4.79 ± 2.91 cm, respectively). The LSD values indicated significant differences among treatments for radicle length, shoot length, and seedling length but not for germination percentage (NS) (Table 7).

In summary, while lower concentrations of antifungal agents (e.g., 1 ppm) promote better overall growth in terms of germination rate, radicle length, shoot length, and seedling length, higher concentrations (e.g., 10,000 ppm) tend to inhibit growth. Hypo treatment showed the most consistent positive effects across all parameters, suggesting its potential as a beneficial treatment to enhance germination and seedling development without significant inhibition of growth.

## 3. Discussion

### 3.1. Germination Experiment

The germination test assessed the effect of various temperatures (5, 10, 15, 20, 25, 30, 35, and 40 °C) on pea seed germination. The results, shown in Figure 1 and Figure 2, indicated that optimal germination occurred between 20 °C and 30 °C, with the highest performance observed at 25 °C. Temperatures below 15 °C and above 35 °C significantly hindered germination rates and uniformity. At 5 °C, the germination process was notably slow, with a maximum germination rate of 77.77% reached after 7.11 days. Conversely, at 40 °C, the germination rate was only 2.22%, reflecting severe thermal stress. The data clearly show that temperature is a critical factor in pea seed germination. Seeds germinated most rapidly and uniformly within the 20–30 °C range, where the mean germination time (MGT) was the shortest, and the Timsons’ Germination Energy Index (GEI) was the highest. For instance, at 25 °C, the MGT was just 2.36 days, and the GEI peaked at 1213.48, indicating vigorous germination. In contrast, extreme temperatures of 5 °C and 40 °C resulted in poor germination performance characterized by longer MGTs, lower GEIs, and higher variability, as illustrated in Figure 2 and Table 1.

These findings align with previous research on the temperature sensitivity of seed germination. Some studies [28,29,30] have similarly highlighted the importance of temperature in determining the duration and success of germination. Additionally, the significant reduction in germination rates at both low and high extremes corresponds with the observations of [31], which noted optimal germination conditions for pea seeds at 10–20 °C. The observed decline in performance at 35 °C and the near failure at 40 °C are consistent with the known effects of thermal stress on seed metabolism and enzyme activity.

Ensuring that pea seeds are sown in environments where temperatures are maintained within the optimal range of 20–30 °C can maximize germination success and seedling vigor. This knowledge is particularly relevant in the context of climate change, where temperature extremes may become more common. However, the controlled laboratory conditions may not fully replicate field environments, where additional factors such as soil moisture, light availability, and microbial interactions can also influence germination. Furthermore, this study focused on a single pea cultivar, and the results may vary with different cultivars or under different environmental conditions.

### 3.2. Initial Development Test

The initial development test investigated the early growth of pea seedlings under different temperature conditions (5, 10, 15, 20, 25, 30, and 35 °C). The results, shown in Figure 4 and Figure 5, indicated that temperatures between 15 °C and 25 °C were optimal for seedling growth, as evidenced by the longest radicle and shoot lengths. At temperatures outside this range, particularly at 5 °C and 35 °C, seedling development was significantly reduced. For instance, at 5 °C, seedlings only reached a mean length of 0.56 cm by the end of the experiment, while at 35 °C, seedling growth was almost negligible. The data suggest that the initial development of pea seedlings is highly temperature-dependent, with the optimal range being 15–25 °C. Within this range, seedlings showed robust growth, with mean radicle lengths of 8.48 ± 4.72 cm at 15 °C, 8.07 ± 4.90 cm at 20 °C, and 7.36 ± 5.03 cm at 25 °C. These findings highlight the importance of maintaining suitable temperature conditions during the early stages of pea cultivation to ensure vigorous seedling development.

Our results align with the existing literature on the temperature sensitivity of seedling growth. Some studies [3,15] have shown that optimal temperature conditions are critical for the metabolic processes involved in seedling growth. The observed decline in seedling performance at 5 °C and 35 °C is consistent with the findings of [14], who noted that extreme temperatures can impair mitochondrial function and reduce energy production necessary for growth.

Additionally, the experiment considered the impact of varying water quantities, measured as multiples of the thousand-kernel weight (TKW), on seedling growth. The optimal water range in a Petri dish with 9 cm of diameter for achieving maximum seedling length was found to be between 8 and 14 milliliters. Even minimal water amounts, such as 2 mL, supported germination. Root growth responded positively to increased water up to a peak at 7 milliliters but declined beyond the optimal level. Both dry weight and TKW measurements consistently indicated an optimal water potential range of 7–11 milliliters for optimal pea growth. These findings are supported by research from [11], which demonstrated that water availability is a critical factor for seedling development, particularly under varying temperature conditions.

By understanding the optimal temperature range and water requirements for pea seedling development, farmers can better plan their planting schedules and manage environmental conditions to maximize growth. This is particularly important in the context of climate change, where temperature fluctuations and water scarcity may impact seedling viability and crop yields.

### 3.3. Effect of Seed Density on Pea

The seed density experiment examined the effect of varying seed densities on the germination and initial development of pea seedlings. The densities tested were five, seven, nine, and eleven seeds per 9 cm Petri dish. The results, displayed in Table 5, indicated that the placement of five pea seeds per Petri dish consistently resulted in the highest proportion of normal seedlings, suggesting optimal germination conditions. Higher densities, particularly 11 seeds per dish, showed greater variability in germination outcomes, including a higher incidence of radicle-only seedlings and short seedlings. The data from Table 5 demonstrate that the aggregated germination values, which account for the combined outcomes of different germination stages, were highest for the 5-seed density (0.960 ± 0.127) and lowest for the 11-seed density (0.652 ± 0.327). This suggests that increased competition for resources such as water, light, and nutrients at higher seed densities adversely affects seedling development. The statistical analysis of variance revealed no significant differences in the germination percentages across different seed densities; however, the quality and uniformity of seedlings were noticeably better at lower densities.

These findings align with previous research highlighting the impact of seed density on plant growth. Some studies [32,33] have shown that higher seed densities can lead to increased competition for limited resources, resulting in reduced seedling growth and greater variability in germination outcomes. This is consistent with the results observed in this experiment, where higher seed densities led to a mix of germination stages, including ungerminated seeds, seeds with only initial growth, and seeds with short or normal seedlings.

The practical implications of this study are significant for both laboratory and field practices. Maintaining lower seed densities can enhance the uniformity and health of pea seedlings, potentially leading to better crop establishment and yield. This knowledge can inform guidelines for optimal seeding strategies, particularly in controlled environments where precision is crucial. Overall, the seed density experiment underscores the importance of optimizing seed density to ensure successful and uniform seedling development.

### 3.4. Effect of Antifungal Treatment on Pea Seeds

The antifungal experiment evaluated the impact of different antifungal treatments on the germination and initial growth of pea seeds. The results indicated no significant differences among the treatments in terms of germination percentage, radicle length, shoot length, and overall seedling length. The Hypo treatment achieved the highest germination percentage (100.0 ± 0.0), followed by the Bordeaux solution (97.78 ± 4.97) and the control (93.33 ± 14.91). However, the differences were not statistically significant, as shown in Table 6. The data suggest that while the Hypo treatment slightly enhanced germination and early seedling growth, the variations among treatments were not sufficient to establish a clear advantage over the control. This indicates that the antifungal treatments applied did not significantly alter the germination outcomes. Previous research like [25,34] has demonstrated that antifungal treatments can influence seedling growth by mitigating fungal infections that impede seed germination. Additionally, the use of antifungal agents like Hypo can have broader implications for agricultural practices. By effectively reducing fungal infections and promoting seedling vigor, these treatments can lead to more robust plant establishment and potentially higher yields. This is particularly relevant in organic farming and sustainable agriculture, where chemical fungicides are limited, and natural or less toxic alternatives are preferred, as the authors of [35] suggested in their study. In conclusion, the antifungal experiment demonstrates that while antifungal treatments like a Hypo and Bordeaux solution can slightly enhance germination and early seedling growth, the differences observed were not statistically significant. These findings suggest that under optimal germination conditions, the benefits of antifungal treatments may be limited. Future research should explore field trials and different cultivars to validate these findings and optimize antifungal treatment strategies for diverse agricultural settings.

## 4. Materials and Methods

This research explored the impact of abiotic stressors such as water and temperature, alongside seedling density and strategies for fungal growth suppression, on the germination and development of pea seedlings (*Pisum sativum* L. ‘Balltrap’) in vitro, employing the ICO105 growth or climate chamber (Memmert GmbH, Schwabach, Germany) for laboratory experiments. The investigation was carried out in 2023 and 2024 at the laboratory of the Institute of Agronomy of the Hungarian University of Agriculture and Life Sciences in Gödöllő and delved into how different levels of temperature, standards of water application, seedling densities, and methods of applying antifungals affect seed germination and seedling growth, as demonstrated through a series of five sub-experiments.

### 4.1. Germination Experiment

Our first sub-experiment was a germination count test of pea seeds. During this phase of the experiment, we explored the onset of seed germination in Petri dishes with filter paper at different temperatures. The temperatures were 5, 10, 15, 20, 25, 30, 35, and 40 °C. We used five replications in every case and nine seeds per Petri dish, with 9 mL of distilled water. The experiment lasted for 14 days at every temperature. We recorded the number of newly germinated seeds every 24 h (Figure 9).

The experiment utilized key metrics based on the Four-parameter hill function by [36] as follows:(1)$$y=y_{0}+\frac{ax^{b}}{x^{b}+c^{b}}$$

The time to maximum germination rate (TMGR) was calculated as partial derivative of the Four-parameter hill function:(2)$$TMGR=\sqrt[{b}]{\frac{c^{b}(b-1)}{b+1}}$$
where *y* is the cumulative germination percentage at time *x*, *y₀* is the intercept on the y-axis, *a* represents the asymptote, *b* is a parameter that controls the shape and steepness of the germination curve, and *c* is the half-maximal activation level.

We calculated the t_50_ value which estimates the time at which 50% of the seeds have germinated, providing an important metric for comparing the speed of germination across different conditions by [37] as follows:(3)$$t_{50}=T_{i}+\left(\frac{\frac{N}{2}-N}{N_{j}-N_{i}}\right)\times (T_{j}{-T}_{i})$$
where *t*_50_ is the time to reach 50% of the final or maximum germination, *N* is the final number of germinated seeds, and *N_i_* and *N_j_* are the cumulative numbers of seeds germinated at times *T_i_* and *T_j_* respectively, with *N_i_ < N/2 < N_j_*.

We determined the mean germination time (MGT) by [38], which indicates how quickly the majority of seeds in a batch terminate as follows:(4)$$MGT=\frac{\sum \left(N_{i}\times T_{i}\right)}{N_{i}}$$
where *N_i_* is the number of seeds germinated at time *T_i_*.

The uniformity of germination (U) shows the difference in time between the maximum and minimum germination times, indicating the uniformity of the germination process. We calculated it as follows:(5)$$U=t_{max}-t_{min}$$
where *t_max_* is the time at which the maximum percentage of seeds has germinated and *t_min_* is the time at which the minimum percentage of seeds has germinated.

The area under the curve (AUC) for the germination rate was calculated based on the Four-parameter hill function also by [36]. Timsons’ Germination Energy Index (GEI) by [39] was calculated as follows:(6)$$\mathrm{GEI}=\sum k=\sum_{i=1}^{k}G_{i}$$
where *G_i_* is the cumulative germination percentage in time interval *i,* and *k* is the total number of time intervals.

The synchronization index (E) by [38] was also calculated to provide a comprehensive assessment of germination performance:(7)$$E=-\sum_{i=1}^{k}f_{i}{log}_{2}f_{i}$$
where *f_i_* is the relative frequency of germination, *N_i_* is the number of seeds germinated on the *i*th time interval, and *k* is the total number of time intervals.

### 4.2. Initial Development Test

This sub-experiment was a continuation of the previous one. Our goal was to investigate the development of pea seedlings under seven distinct temperature conditions, namely 5, 10, 15, 20, 25, 30, and 35 °C. We excluded the 40 °C treatment based on the result of our first sub-experiment. The Petri dishes, 9 cm in diameter, were labeled, uniformly lined with filter paper, and filled with nine pea seeds and 9 mL of distilled water for each, with five replications to maintain consistency. The non-germinated seeds were tallied, and the lengths of all the seedlings’ radicles and plumules were measured. The measurement process commenced when approximately 80% of the seedlings in the Petri dishes had reached a length of 0.5 cm. Once a day, four Petri dishes were taken out from the growth chamber from each temperature to take physical measurements of the seedlings. Later, germination and development percentages were determined.

### 4.3. Water Amount Experiment

This sub-experiment had two parts. Firstly, in sterile Petri dishes lined with a single sterile filter paper, 9-9 pea seeds were subjected to 15 different distilled water amounts (0–14 mL by 1 mL steps). In the second part, based on milliliter intervals, 19 amounts of distilled water were applied as the multiple percentages of thousand-kernel weight (TKW) of the seeds. TKW is a measure of the physical property of seeds, and it denotes the weight of one-thousand seeds from a given batch, expressed in grams. TKW differs strongly in the case of different pea species, where extremities can be 150–600 g. With this % method, different species can be compared. The used pea seeds had a TKW of 515.5 g. To determine the minimum TKW percentage, we used the following Equation (8), based on our previous experiments [25,27]:(8)$${WV}_{1\%}=\frac{TKW\times n}{100000}=\frac{515.5g\times 9}{100000}=0.0464\;\mathrm{mL}$$
where *WV*_1%_ represents 1% of the proposed water amount calculated in milliliters, *TKW* is the weight of a thousand pea seeds in grams, and *n* denotes the number of pea seeds per Petri dish.

The Petri dishes (PD) were labeled, and nine seeds were uniformly placed in each, with five replications for each of the 34 treatments. Following a 10-day incubation period, physical measurements and assessments of the radicles and shoots were conducted, and the quantity of non-germinated seeds was measured. These radicles and shoots were tagged afterward and dried in an oven at 65 °C for two days to obtain the dry weight. The dry weights of the radicles and plumules of nine seedlings from each experimental unit were then recorded using an analytical scale.

### 4.4. Density Experiment

The effect of seed density on germination was investigated as the number of seeds can impact seedling development. For this sub-experiment, 50 Petri dishes lined with filter paper were used, 10 for each planned density. The germination process was monitored using four different seed quantities, which included placing five, seven, nine, ten, and eleven seeds into separate Petri dishes with the application of an equal amount of 9 milliliters of distilled water for each. In climate chambers, constant temperature was maintained at 20 °C for 10 days. To decrease experimental error and improve the overall precision of the data, 10 repetitions were conducted for each treatment group. After incubation, seedlings were taken out of the chamber for physical measurement, where the measurable parameters were categorized into five groups: seeds that have not yet germinated, seeds where germination has already started, germinated seeds with radicle only, seedlings with a short plumule, and seeds with normal-sized roots and shoots (approximately 3 cm in length). Creating these categories was expedient because it allowed us the opportunity to calculate an aggregated germination value (AGG) that provides useful information about the optimal pea seed density per Petri dish. Based on our previous studies [24], we used the following Equation (9):(9)$$AGG\left(x\right)=\frac{\left(\left(NO-G\times 0\right)+\left(S\times 0.1\right)+\left(R\times 0.25\right)+\left(SP\times 0.65\right)+\left(NP\times 1\right)\right)}{N}$$
where *AGG* is the aggregated germination value; *NO-G* is the non-germinated seed number, S is the number of seeds where germination started, *R* is the number of germinated seeds with radicle only, *SP* is the number of seedlings with a short plumule, *NS* is the number of normal seedlings (*NS*s), and *N* is the total number of the tested seeds.

### 4.5. Antifungal Experiment

Two antifungal experiments were conducted in which Hypo and Bordeaux’s mixture (BM) was applied to pea seeds. Firstly, the seeds were sterilized in separate solutions of 1000 ppm of Bordeaux’s mixture and 3% Hypo for three minutes; after which, they were rinsed with distilled water. In the second run, four different concentrations of the BM fungicide (0, 1, 10, 100, 1000, and 10,000 ppm) were applied to the growth media. From the two sets, ten-ten replications were incubated at each temperature of 20 °C. After 10 identic days of the incubation period, physical measurements and evaluations were carried out, where the Petri dishes were removed from the growth chambers, and the lengths of the radicle and plumule were measured for all seedlings, while the count of germinated seeds was recorded.

### 4.6. Statistical Analysis

MS Excel 365 was used to tabulate the data. For the data normality verifications, Kolmogorov–Smirnov and Shapiro–Wilk tests were conducted, and an analysis of variance (ANOVA) and Fisher’s test of least significant differences (LSDs) were conducted. Where the normality assumption was not reached, we used non-parametric tests such as Kruskal–Wallis with Bonferroni’s post hoc test and the Mann–Whitney U test for pairwise comparisons. For statistical analysis, we used the IBM SPSS V27 (New York, NY, USA) and RStudio V2024.04.2 (Rstudio Team, 2024) software. Germination indices were calculated, and fitting curves were created with the help of several R packages for data analysis and visualization, including ‘germination metrics’ [40], ‘dplyr’ [41], ‘tidyr’ [42], ‘ggplot2’ [43], and ‘patchwork’ [44]. The effects of the water level, seed number, and antifungal treatment on the germination percentage, radicle length, shoot length, and seedling growth were analyzed and a sigmoid curve model was applied in Python using various Python packages for data analysis and visualization, including Pandas [45], NumPy [46], Matplotlib [47], Seaborn [48], and Scikit-learn [49].

## 5. Conclusions

This study offers a detailed analysis of the optimal conditions for pea (*Pisum sativum* L.) germination, focusing on key factors such as water levels, temperature, seed density, and the effects of antifungal treatments in a laboratory setting. The research aimed to determine the best conditions to maximize germination rates and enhance seedling vigor, which are crucial for research, breeding, and microgreen production.

For the pea variety ‘Balltrap’, the optimal germination occurred within a temperature range of 15 °C to 25 °C. Germination rates decreased significantly at temperatures below 15 °C and above 25 °C, with no germination observed at 35 °C. This finding underscores the importance of maintaining moderate temperatures for successful germination and seedling growth, particularly in the context of potential temperature extremes due to climate change.

Water levels between 8 and 14 mL per Petri dish were found to be ideal for promoting germination and seedling development. Specifically, water amounts ranging from 7 to 11 mL consistently supported strong root and shoot growth. Minimal water levels, such as 2 mL, were enough to initiate germination but could not sustain further growth, while excessive water beyond 14 mL negatively impacted seedling development. This highlights the critical role of precise water management in ensuring healthy germination and growth.

The study also found that seed density significantly affects germination outcomes. Placing five seeds per 9 cm Petri dish was identified as the optimal density for healthy shoot and root growth. Higher densities, such as seven or nine seeds per dish, led to increased competition for resources, resulting in variable germination rates and reduced seedling quality. This insight is crucial for optimizing laboratory and field practices to achieve uniform and vigorous seedling development.

Regarding antifungal treatments, the study found that treatments like the Hypo and Bordeaux mixture provided slight improvements in germination and early seedling growth. However, these improvements were not statistically significant compared to the control, and higher concentrations of antifungal agents did not yield significant benefits. This suggests that while antifungal treatments can offer some advantages, their application should be carefully considered and tailored to specific environmental conditions.

Overall, this research provides practical guidelines for improving germination rates and seedling vigor, with important implications for sustainable agriculture. The study’s findings are particularly relevant for researchers, breeders, and microgreen producers, offering valuable insights into optimizing germination in laboratory conditions. Future research should explore these factors in different field settings and across various pea cultivars to refine these recommendations further.

## Figures and Tables

**Figure 1 plants-13-02776-f001:**
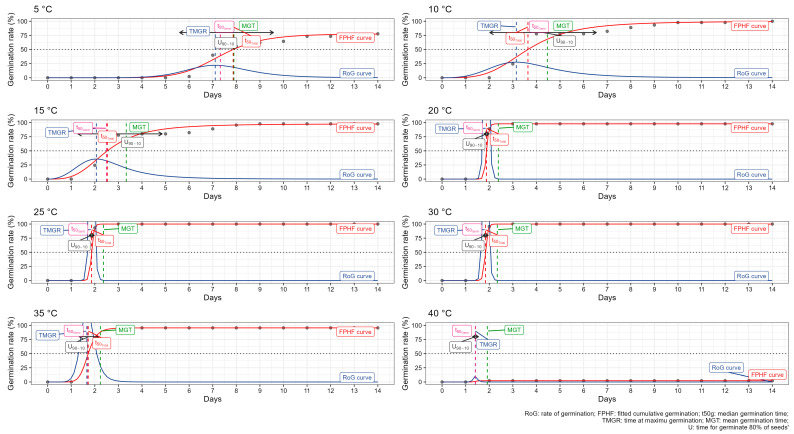
Germination data of pea (*Pisum sativum* L.) at different temperatures. RoG (blue continuous): rate of germination; FPHF (red continuous): fitted cumulative germination; t_50_g (pink dashed): median germination time; TMGR (blue dashed): time at maximum germination; MGT (green dashed): mean germination time; U (black dashed): time for germinating 80% of seeds.

**Figure 2 plants-13-02776-f002:**
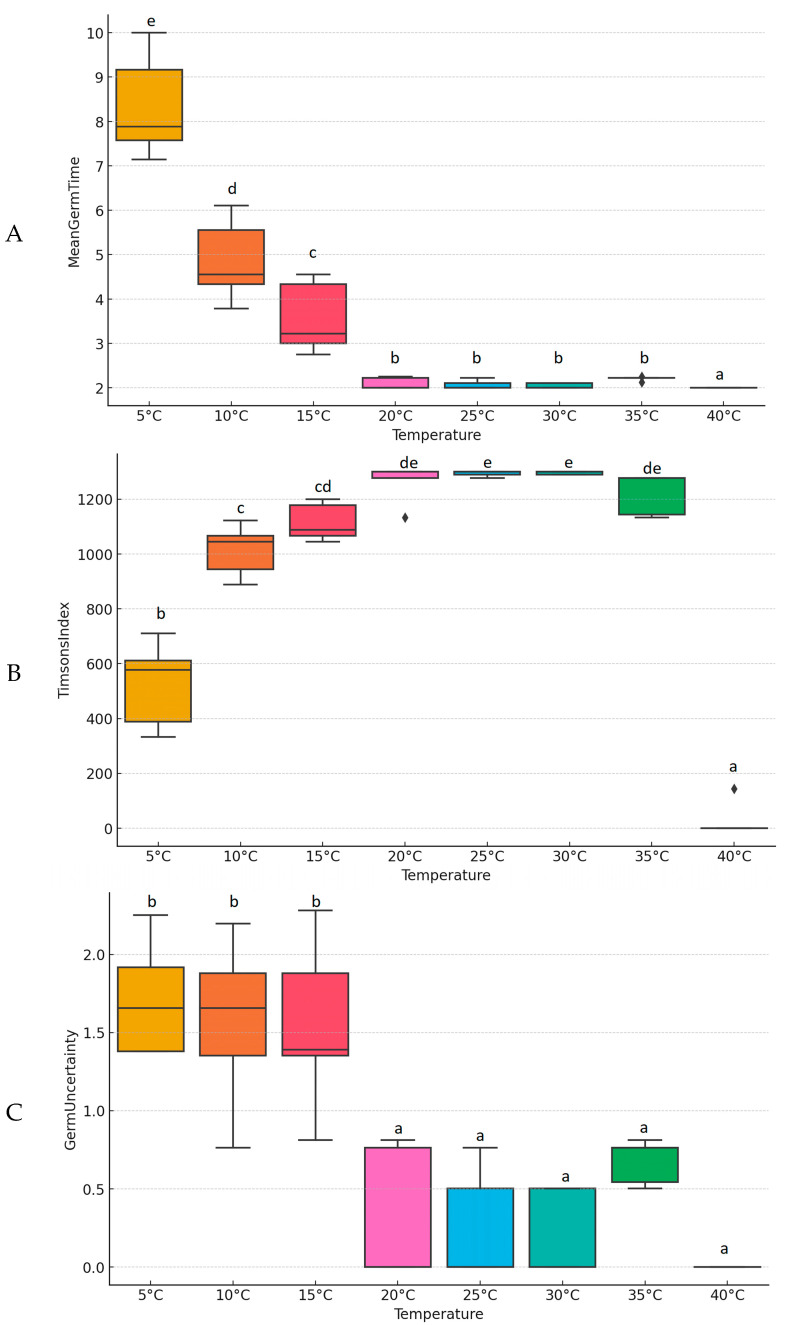
(**A**) Mean germination time (MGT), (**B**) Timsons’ Germination Energy Index (GEI), (**C**) synchronization index (Z) of pea by different temperatures. Different lowercase letters indicate statistically significant disparities in the means at *p* < 0.05 based on Tukey’s HSD post hoc test.

**Figure 3 plants-13-02776-f003:**
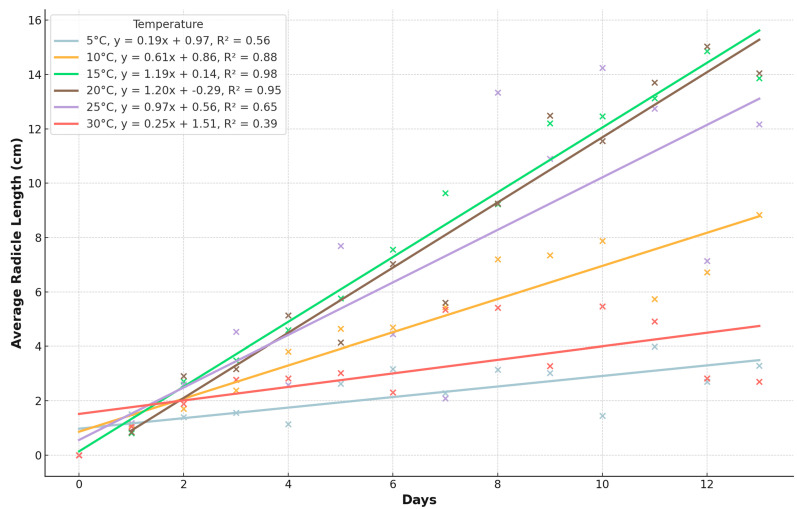
The growth of pea seeds’ radicle length over time at different temperatures.

**Figure 4 plants-13-02776-f004:**
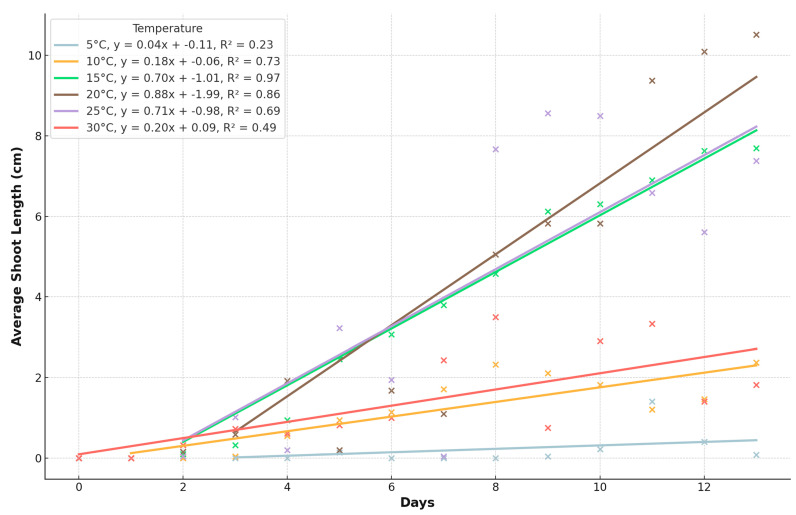
The growth of shoots of pea seeds over time at different temperatures.

**Figure 5 plants-13-02776-f005:**
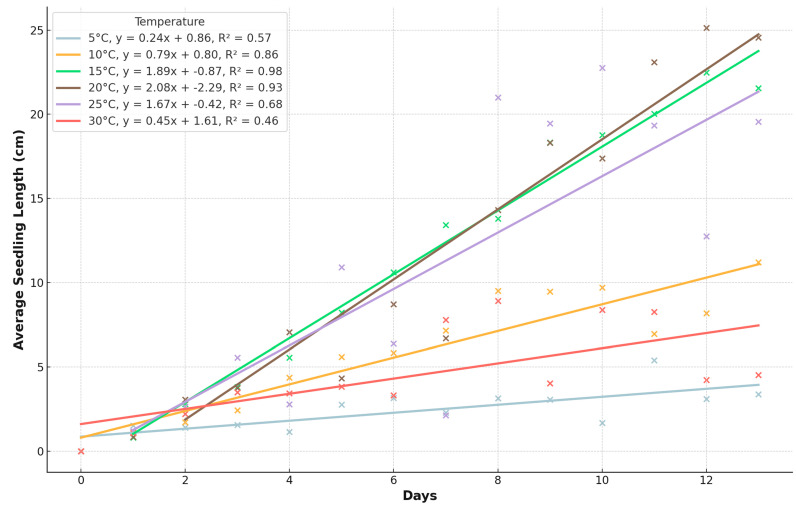
The trend in seedling length growth over time at different temperatures.

**Figure 6 plants-13-02776-f006:**
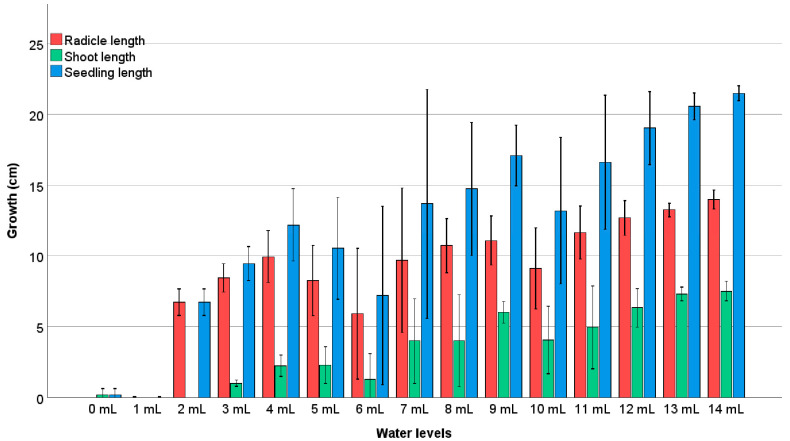
Pea seeds’ radicles, shoots, and seedlings respond to the various water availability levels and growth reaction to water levels of 1-milliliter intervals (0–14 mL).

**Figure 7 plants-13-02776-f007:**
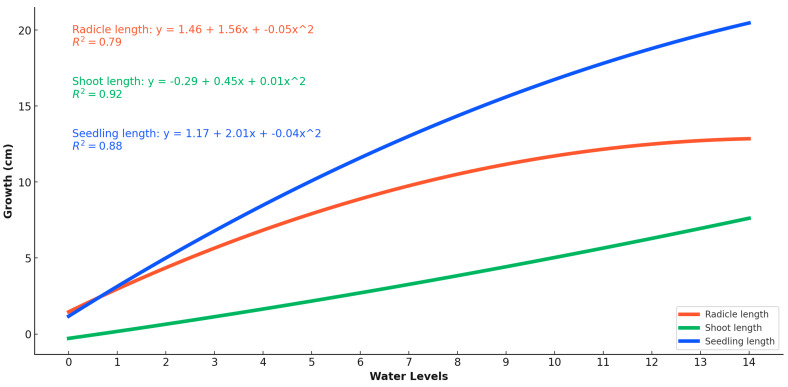
Pea seeds’ radicles, shoots, and seedlings respond to the various water availability levels and plant part length growth as a reaction to water levels of 1-milliliter intervals (0–14 mL).

**Figure 8 plants-13-02776-f008:**
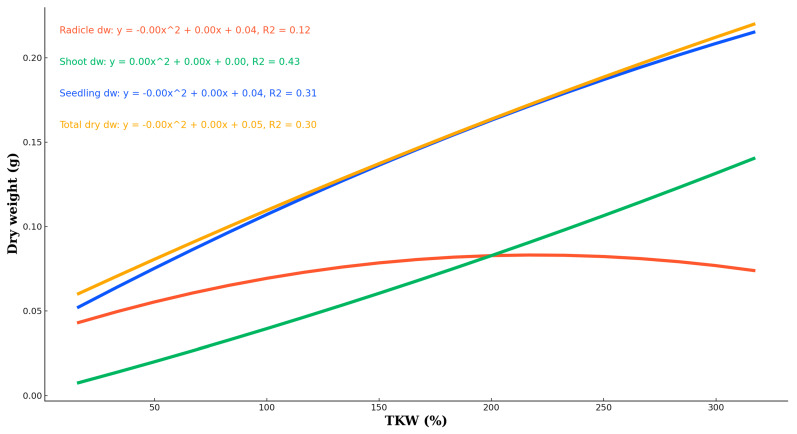
Pea seeds’ radicles, shoots, and seedlings respond to the various water availability levels and dry weight accumulation as a reaction to various water levels supplied as a percentage correlated to the TKW.

**Figure 9 plants-13-02776-f009:**
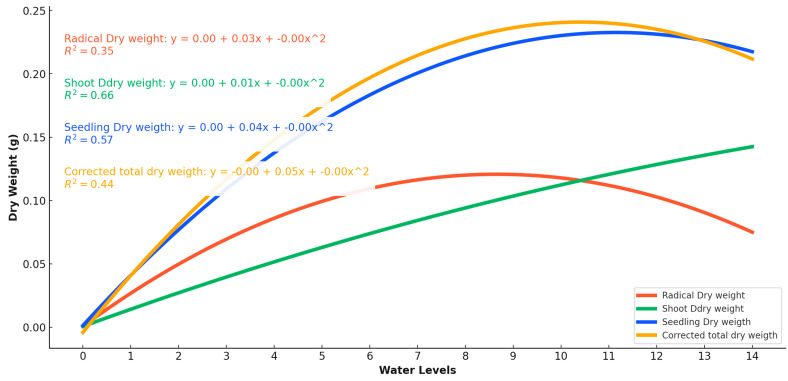
Pea seeds’ radicles, shoots, and seedlings respond to the various water availability levels and dry weight accumulation under various water levels of 1-milliliter intervals (0–14 mL).

**Table 1 plants-13-02776-t001:** Results of Bonferroni post hoc test of three selected germination indices of pea at different temperatures.

Mean Germ. Time	**Temperature**	**5 °C**	**10 °C**	**15 °C**	**20 °C**	**25 °C**	**30 °C**	**35 °C**	**40 °C**
	5 °C	-							
	10 °C	−5.8	-						
	15 °C	−9.2	3.4	-					
	20 °C	−23.20 *	17.40 *	14	-				
	25 °C	−24.90 *	19.10 *	15.70 *	1.7	-			
	30 °C	−25.90 *	20.10 *	16.70 *	2.7	1	-		
	35 °C	−16.90 *	11.1	7.7	−6.3	−8	−9	-	
	40 °C	−28.50 *	22.7	19.3	5.3	3.6	2.6	11.6	-
Timsons’ Index	**Temperature**	**5 °C**	**10 °C**	**15 °C**	**20 °C**	**25 °C**	**30 °C**	**35 °C**	**40 °C**
	5 °C	-							
	10 °C	6	-						
	15 °C	10.2	−4.2	-					
	20 °C	22.70 *	−16.70 *	−12.5	-				
	25 °C	24.80 *	−18.80 *	−14.6	−2.1	-			
	30 °C	25.60 *	−19.60 *	−15.4	−2.9	−0.8	-		
	35 °C	15.70 *	−9.7	−5.5	7	9.1	9.9	-	
	40 °C	−5	11	15.2	27.70 *	29.80 *	30.60 *	20.70 *	-
Synchronization index	**Temperature**	**5 °C**	**10 °C**	**15 °C**	**20 °C**	**25 °C**	**30 °C**	**35 °C**	**40 °C**
	5 °C	-							
	10 °C	−2.6	-						
	15 °C	−1.7	−0.9	-					
	20 °C	−20	17.4	18.3	-				
	25 °C	−21.90 *	19.3	20.2	1.9	-			
	30 °C	−23.00 *	20.4	21.3	3	1.1	-		
	35 °C	−13	10.4	11.3	−7	−8.9	−10	-	
	40 °C	−26.60 *	24.00 *	24.90 *	6.6	4.7	3.6	13.6	-

Each row tests the null hypothesis that the Sample 1 and Sample 2 distributions are the same. Asymptotic significances (2-sided tests) are displayed. * The results are significant at *p* = 0.05.

**Table 2 plants-13-02776-t002:** Pairwise Mann–Whitney U test results for seedling lengths across different temperatures of pea seeds.

Temperature	5 °C	10 °C	15 °C	20 °C	25 °C	30 °C
5 °C	-	U = 510, *p* = 0.000 *	U = 336, *p* = 0.000 *	U = 345, *p* = 0.000 *	U = 604, *p* = 0.000 *	U = 844, *p* = 0.001 *
10 °C		-	U = 753, *p* = 0.000 *	U = 928, *p* = 0.006 *	U = 1020, *p* = 0.031 *	U = 1681, *p* = 0.033 *
15 °C			-	U = 1374, *p* = 0.886	U = 1441, *p* = 0.565	U = 2126, *p* = 0.000 *
20 °C				-	U = 1501, *p* = 0.333 *	U = 2018, *p* = 0.000 *
25 °C					-	U = 1852, *p* = 0.001 *
30 °C						-

* Results are significant at level *p* = 0.05.

**Table 3 plants-13-02776-t003:** Germination and seedling measured traits of *Pisum sativum* L. seeds reacting to water level application on scale of 1 mL water intervals.

Water mL	Germination (%)	Radicle Length (cm)	Shoot Length (cm)	Seedling Length (cm)	Radicle DW (g)	Shoot DW (g)	Seedling DW (g)	Corrected DW (g)
0	0.0 ± 0.0 a	0.0 ± 0.0 a	0.0 ± 0.0 a	0.0 ± 0.0 a	0.0 ± 0.0 a	0.0 ± 0.0 a	0.0 ± 0.0 a	0.0 ± 0.0 a
1	2.2 ± 5.0 a	0.0 ± 0.0 a	0.0 ± 0.0 a	0.0 ± 0.0 a	0.0 ± 0.0 a	0.0 ± 0.0 a	0.0 ± 0.0 a	0.0 ± 0.0 a
2	97.8 ± 5.0 b	6.7 ± 0.9 bc	0.0 ± 0.0 a	6.7 ± 0.9 ab	0.072 ± 0.005 ab	0.0 ± 0.0 a	0.072 ± 0.005 ab	0.074 ± 0.007 a
3	93.3 ± 9.9 b	8.5 ± 1.0 bcde	1.0 ± 0.2 ab	9.5 ± 1.2 bcd	0.069 ± 0.01 ab	0.028 ± 0.013 ab	0.097 ± 0.011 ab	0.105 ± 0.016 a
4	100.0 ± 0.0 b	9.9 ± 1.8 bcdef	2.3 ± 0.8 abc	12.2 ± 2.6 bcde	0.1 ± 0.011 ab	0.158 ± 0.208 b	0.258 ± 0.21 bc	0.258 ± 0.21 ab
5	97.8 ± 5.0 b	8.2 ± 2.5 bcd	2.3 ± 1.3 abc	10.5 ± 3.6 bcd	0.073 ± 0.022 ab	0.065 ± 0.028 ab	0.138 ± 0.048 abc	0.143 ± 0.057 a
6	75.6 ± 43.3 b	5.9 ± 4.6 b	1.3 ± 1.8 ab	7.2 ± 6.3 abc	0.063 ± 0.036 ab	0.038 ± 0.053 ab	0.101 ± 0.088 ab	0.067 ± 0.088 a
7	84.4 ± 29.0 b	9.7 ± 5.1 bcdef	4.0 ± 3.0 bcd	13.7 ± 8.1 bcdef	0.273 ± 0.397 b	0.087 ± 0.066 ab	0.36 ± 0.355 c	0.472 ± 0.402 b
8	97.8 ± 5.0 b	10.7 ± 1.9 bcdef	4.0 ± 3.3 bcd	14.7 ± 4.7 cdefg	0.092 ± 0.02 ab	0.079 ± 0.058 ab	0.17 ± 0.069 abc	0.173 ± 0.067 a
9	100.0 ± 0.0 b	11.1 ± 1.7 cdef	6.0 ± 0.8 d	17.1 ± 2.1 defg	0.106 ± 0.014 ab	0.113 ± 0.017 ab	0.219 ± 0.024 abc	0.219 ± 0.024 ab
10	100.0 ± 0.0 b	9.1 ± 2.9 bcdef	4.1 ± 2.4 bcd	13.2 ± 5.2 bcdef	0.069 ± 0.024 ab	0.09 ± 0.044 ab	0.16 ± 0.066 abc	0.16 ± 0.066 a
11	97.8 ± 5.0 b	11.7 ± 1.9 def	5.0 ± 2.9 cd	16.6 ± 4.7 defg	0.084 ± 0.022 ab	0.103 ± 0.058 ab	0.187 ± 0.08 abc	0.188 ± 0.077 a
12	100.0 ± 0.0 b	12.7 ± 1.2 def	6.3 ± 1.4 d	19.0 ± 2.6 efg	0.092 ± 0.013 ab	0.129 ± 0.027 ab	0.222 ± 0.035 abc	0.222 ± 0.035 ab
13	100.0 ± 0.0 b	13.2 ± 0.5 ef	7.3 ± 0.5 d	20.6 ± 0.9 fg	0.102 ± 0.021 ab	0.151 ± 0.009 b	0.253 ± 0.024 abc	0.253 ± 0.024 ab
14	100.0 ± 0.0 b	14.0 ± 0.7 f	7.5 ± 0.7 d	21.5 ± 0.5 g	0.101 ± 0.011 ab	0.15 ± 0.012 b	0.251 ± 0.009 abc	0.251 ± 0.009 ab
LSD	10.17	2.94	2.74	5.58	0.035	0.049	0.077	0.084

Different lowercase letters in columns indicate statistically significant disparities in the means by Tukey HSD at *p* < 0.05. LSD values are calculated at *p* < 0.05. DW means the dry weight of the radicle, shoot, or seedlings. Corrected DW shows the mean of the adjusted dry weight, which excludes the non-germinated seedlings.

**Table 4 plants-13-02776-t004:** Germination and seedling measured traits of *Pisum sativum* L. seeds reacting to water level application in correlation to TKW.

Treatment (mL)	TKW (%)	Germination (%)	Radicle Length (cm)	Shoot Length (cm)	Seedling Length (cm)	Radicle DW (g)	Shoot DW (g)	Seedling DW (g)	Corrected DW (g)
0.75	16.17	0.0 ± 0.0 a	0.0 ± 0.0 a	0.0 ± 0.0 a	0.0 ± 0.0 a	0.0 ± 0.0 a	0.0 ± 0.0 a	0.0 ± 0.0 a	0.0 ± 0.0 a
1.55	33.41	46.67 ± 14.49 b	1.49 ± 0.49 ab	0.0 ± 0.0 a	1.06 ± 0.67 a	0.02 ± 0.0 ab	0.0 ± 0.0 a	0.02 ± 0.0 a	0.04 ± 0.0 a
2.30	49.57	97.78 ± 4.97 c	7.47 ± 0.67 cdefg	0.02 ± 0.05 a	7.49 ± 0.7 abcd	0.08 ± 0.0 cde	0.0 ± 0.0 a	0.08 ± 0.01 abc	0.08 ± 0.01 abc
3.10	66.82	97.78 ± 4.97 c	8.79 ± 2.12 cdefg	1.52 ± 2.56 abc	10.31 ± 1.33 bcde	0.08 ± 0.01 cde	0.01 ± 0.01 ab	0.09 ± 0.02 abcd	0.1 ± 0.02 abcd
3.85	82.98	100.0 ± 0.0 c	11.6 ± 1.4 efg	2.6 ± 0.85 abc	14.2 ± 1.73 cde	0.11 ± 0.01 e	0.06 ± 0.02 abcde	0.17 ± 0.02 bcde	0.17 ± 0.02 bcde
4.65	100.23	97.78 ± 4.97 c	10.69 ± 1.47 defg	3.84 ± 0.51 abcdef	14.53 ± 1.62 cde	0.1 ± 0.01 de	0.09 ± 0.01 bcdef	0.19 ± 0.02 cde	0.19 ± 0.01 cde
5.40	116.39	100.0 ± 0.0 c	12.41 ± 0.65 g	4.54 ± 0.65 bcdef	16.96 ± 1.23 de	0.11 ± 0.02 e	0.1 ± 0.02 cdef	0.22 ± 0.03 e	0.22 ± 0.03 e
6.20	133.64	95.56 ± 6.09 c	4.16 ± 2.31 abc	0.59 ± 0.92 ab	4.74 ± 3.23 ab	0.05 ± 0.02 bc	0.02 ± 0.03 ab	0.07 ± 0.04 ab	0.07 ± 0.04 ab
6.95	149.80	100.0 ± 0.0 c	9.16 ± 2.6 cdefg	3.02 ± 1.14 abcd	12.18 ± 3.45 bcde	0.08 ± 0.02 cde	0.08 ± 0.02 bcdef	0.16 ± 0.04 bcde	0.16 ± 0.04 bcde
7.75	167.04	100.0 ± 0.0 c	8.48 ± 2.98 cdefg	3.21 ± 1.92 abcde	11.69 ± 4.89 bcde	0.08 ± 0.04 cde	0.08 ± 0.05 bcdef	0.16 ± 0.09 bcde	0.16 ± 0.09 bcde
8.50	183.21	100.0 ± 0.0 c	5.87 ± 3.72 bcde	1.57 ± 2.73 abc	7.43 ± 6.42 abcd	0.05 ± 0.04 bc	0.02 ± 0.02 ab	0.04 ± 0.01 a	0.04 ± 0.01 a
9.30	200.45	93.33 ± 9.94 c	5.19 ± 4.8 bcd	1.47 ± 2.7 abc	6.66 ± 7.49 abc	0.05 ± 0.04 bc	0.03 ± 0.06 abc	0.08 ± 0.1 abc	0.08 ± 0.09 abc
10.05	216.62	95.56 ± 6.09 c	6.52 ± 4.24 bcdef	1.99 ± 3.18 abc	8.51 ± 7.38 abcd	0.06 ± 0.03 bcd	0.04 ± 0.06 abcd	0.1 ± 0.09 abcd	0.1 ± 0.09 abcd
10.80	232.78	97.78 ± 4.97 c	10.13 ± 4.25 defg	5.1 ± 3.16 cdef	15.23 ± 7.33 cde	0.07 ± 0.03 cde	0.11 ± 0.06 cdef	0.18 ± 0.09 cde	0.18 ± 0.09 cde
11.60	250.03	100.0 ± 0.0 c	11.41 ± 1.24 efg	7.09 ± 1.51 def	18.5 ± 2.54 e	0.09 ± 0.02 cde	0.14 ± 0.03 f	0.23 ± 0.05 e	0.23 ± 0.05 e
12.35	266.19	97.78 ± 4.97 c	11.87 ± 1.91 fg	7.57 ± 1.2 f	19.43 ± 3.08 e	0.09 ± 0.03 cde	0.15 ± 0.02 f	0.24 ± 0.04 e	0.24 ± 0.04 e
13.15	283.44	97.78 ± 4.97 c	9.97 ± 2.47 defg	5.49 ± 3.38 cdef	15.46 ± 5.47 cde	0.07 ± 0.02 cde	0.13 ± 0.04 ef	0.2 ± 0.05 de	0.21 ± 0.06 de
13.90	299.60	100.0 ± 0.0 c	12.21 ± 1.39 fg	7.43 ± 0.85 ef	19.64 ± 2.01 e	0.1 ± 0.01 cde	0.15 ± 0.01 f	0.25 ± 0.02 e	0.25 ± 0.02 e
14.70	316.84	97.78 ± 4.97 c	10.82 ± 3.52 defg	5.56 ± 3.2 cdef	16.38 ± 6.59 de	0.08 ± 0.03 cde	0.11 ± 0.07 def	0.2 ± 0.09 de	0.2 ± 0.08 de
LSD		1.12	0.54	0.41	0.89	0.005	0.007	0.011	0.011

Different lowercase letters in columns indicate statistically significant disparities in the means by Tukey HSD at *p* < 0.05. LSD values are calculated at *p* < 0.05. DW means the dry weight of the radicle, shoot, or seedlings. Corrected DW shows the mean of the adjusted dry weight, which excludes the non-germinated seedlings.

**Table 5 plants-13-02776-t005:** The density-dependent seed germination and seedling development of pea (*Pisum sativum* L.) per Petri dish.

Seed No.	Inactive	Initial Growth	Radicle Only	Short Seedlings	Normal Seedlings	Aggregated Value
5	0.000 ± 0.000	0.000 ± 0.000	0.060 ± 0.190	0.000 ± 0.000	0.940 ± 0.190	0.960 ± 0.127
7	0.014 ± 0.045	0.000 ± 0.000	0.171 ± 0.368	0.029 ± 0.060	0.786 ± 0.358	0.861 ± 0.241
9	0.000 ± 0.000	0.011 ± 0.035	0.200 ± 0.370	0.056 ± 0.108	0.733 ± 0.410	0.838 ± 0.271
11	0.009 ± 0.029	0.018 ± 0.038	0.436 ± 0.443	0.036 ± 0.088	0.482 ± 0.462	0.652 ± 0.327
LSD	0.06 NS	0.059 NS	0.799 NS	0.171 NS	0.831 NS	0.567 NS

NS refers to non-significant differences in columns among the values of the means at *p* < 0.05 level. Short shoot seedlings determined with less than relatively 3 cm in length. The aggregated value based on Equation (2) of the sub-classed fife groups: number of inactive seeds, seeds that initiated germination, only radicle-bearing seedlings, short shoot seedlings, and regular shoot length seedlings.

**Table 6 plants-13-02776-t006:** The growth response of germination, radicles, shoots, and seedlings to different seed priming methods.

Treatment	Germination %	Radicle Length (cm)	Shoot Length (cm)	Seedling Length (cm)
Bordeaux mixture	97.78 ± 4.97 a	5.97 ± 1.29 a	3.24 ± 1.86 a	9.2 ± 2.86 a
Hypo	100.0 ± 0.0 a	9.88 ± 1.02 a	5.24 ± 1.51 a	15.12 ± 2.37 a
Control	93.33 ± 14.91 a	9.11 ± 5.27 a	4.89 ± 2.83 a	14.0 ± 7.95 a
LSD	12.5 NS	4.39 NS	2.95 NS	6.98 NS

NS refers to non-significant differences in columns among the values of the means at the *p* < 0.05 level. Different lowercase letters in columns indicate statistically significant disparities in the means by Tukey HSD at *p* < 0.05.

**Table 7 plants-13-02776-t007:** The growth response of germination, radicles, shoots, and seedlings to varying concentrations of the fungicide Bordeaux mixture.

Treatment	Germination%	Radicle Length (cm)	Shoot Length (cm)	Seedling Length (cm)
1 ppm	100.0 ± 0.0 a	9.57 ± 0.95 c	7.49 ± 2.23 c	17.06 ± 1.48 a
10 ppm	95.56 ± 9.94 a	7.26 ± 3.5 bc	4.84 ± 1.38 bc	12.1 ± 4.75 ab
100 ppm	97.78 ± 4.97 a	3.49 ± 0.83 ab	1.02 ± 1.33 a	4.51 ± 2.09 b
1000 ppm	95.56 ± 6.09 a	3.62 ± 0.37 ab	0.92 ± 1.23 a	4.54 ± 1.29 b
10,000 ppm	84.45 ± 14.91 a	1.77 ± 0.66 a	3.02 ± 2.56 ab	4.79 ± 2.91 b
Control	93.33 ± 14.91 a	9.11 ± 5.27 c	4.89 ± 2.83 bc	14.0 ± 7.95 a
LSD	13.11 NS	3.46	2.65	5.39

NS refers to non-significant differences in columns among the values of the means at *p* < 0.05 level. Different lowercase letters in columns indicate statistically significant disparities in the means by Tukey HSD at *p* < 0.05.

## Data Availability

All data are available on request to the corresponding author.

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
