# Peer review of "Optimizing Water, Temperature, and Density Conditions for In Vitro Pea (Pisum sativum L.) Germination"

_plants, 2024, doi:10.3390/plants13192776_

Round 1

Reviewer 1 Report

Comments and Suggestions for Authors

Comments for authors

1. Line 33-35, the aim of this study could be better set at the end of the introduction section.

2. The Introduction should be reconceived. For example, the first and second paragraphs of the introduction should be integrated into one paragraph, and this part should be more concise.

3. Line 134-138, this should be placed in a Materials and Methods section. The authors should add information on the formula used to assess seed germination.

4. The resolution in all figures is too low to be easily readable, please increase resolution.

5. In the text, there should be no spaces between numbers and units "℃". Milli-litres "ml" should be written as "mL".

6. The description of the results is too long.

7. The color of the lines in Fig 1 are not explained.

8. Please separate each small Figs (a, b, c, ,,,) in Fig 2..

9. In Table 4, Radicle, Shoot and Seedling should be changed to Radicle length, Shoot length and Seedling length, respectively.

10. Could significance tests be added to Fig 2, 7 and 8. Figure 7 and Table 6, Figure 8 and Table 7, seem to duplicate each other?? There is no need to repeat results from tables and figures.

11. “Results indicated that optimal germination occurred between 15 °C and 25 °C, with 17

peak performance at 25 °C. Water levels between 7-11 ml per 9 cm diameter Petri dish supported robust root and shoot development, while minimal water levels initiated germination but did not sustain growth. Five seeds per Petri dish were optimal for healthy development, whereas higher

densities led to increased competition and variable outcomes.” Has this optimum condition been used with success in other pea varieties.

12. The Conclusions part is too long, need to simplify.

Author Response

Dear Reviewr 1, 

Enclosed please find our answer for your revision. 

Reviewer 2 Report

Comments and Suggestions for Authors

The manuscript titled "Optimizing Water, Temperature, and Density Conditions for in vitro Pea (Pisum sativum L.) Germination" proposed to evaluate the effects of temperature, humidity, seed preparation, and antifungal treatments to increase pea germination in laboratories and fields. On the other hand, this manuscript suffers from some deficits as summarized below.

The study cannot be replicated in field conditions as the authors report in the objective. The study focused on laboratory conditions where the environment is controlled. Furthermore, the authors do not really formulate any specific mechanistic hypotheses, they only show general data on pea germination. I'm sorry, but this article does not advance knowledge.

The figures are of low quality, they even give the impression that they are screenshots. I'm sorry, but the data is strange; see in table 3 that where there was no addition of water and with zero germination, there was a Shoot (cm) equal to 0.2 ± 0.4. This does not make sense.

Author Response

Dear Reviewer 2,

Enclosed please find our answer for our reveiew. 

Round 2

Reviewer 1 Report

Comments and Suggestions for Authors

The resolution of the figures looks still low, please increase resolution.

Author Response

Dear Professor,

Enclosed please find our answer for your review. We improved all of the graphs quality. 

Sincerely yours,
The Authors

Reviewer 2 Report

Comments and Suggestions for Authors

The authors have made drastic changes to the manuscript to improve the quality of the writing. However, I still suggest removing Figure 7. This Figure is not necessary.

Author Response

Dear Professor,

Enclosed please find out answers. We deleted the figure as you requested and to improve the quality of the figures we separated the multi-faced figures. 

Sincerely yours,
The Authors
